# Representing Partial Programs with Blended Abstract Semantics

**Maxwell Nye**[*]      **Yewen Pu**         **Matthew Bowers**
**Jacob Andreas**    **Joshua B. Tenenbaum**    **Armando Solar-Lezama**
Massachusetts Institute of Technology

## Abstract

Synthesizing programs from examples requires searching over a vast, combinatorial space of possible programs. In this search process, a key challenge is representing the behavior of a partially written program before it can be executed, to judge if it is on the right track and predict where to search next. We introduce a general technique for representing partially written programs in a program synthesis engine. We take inspiration from the technique of abstract interpretation, in which an approximate execution model is used to determine if an unfinished program will eventually satisfy a goal specification. Here we *learn* an approximate execution model implemented as a modular neural network. By constructing compositional program representations that implicitly encode the interpretation semantics of the underlying programming language, we can represent partial programs using a flexible combination of concrete execution state and learned neural representations, using the learned approximate semantics when concrete semantics are not known (in unfinished parts of the program). We show that these hybrid neuro-symbolic representations enable execution-guided synthesizers to use more powerful language constructs, such as loops and higher-order functions, and can be used to synthesize programs more accurately for a given search budget than pure neural approaches in several domains.

## 1 Introduction

Inductive program synthesis – the problem of inferring programs from examples – offers the promise of building machine learning systems that are interpretable, generalize quickly, and allow us automate software engineering tasks. In recent years, neurally-guided program synthesis, which uses deep learning to guide search over the space of possible programs, has emerged as a promising approach (Balog et al., 2016; Devlin et al., 2017). In this framework, partially-constructed programs are judged to determine if they are on the right track and to predict where to search next. A key challenge in neural program synthesis is *representing* the behavior of partially written programs, in order to make these judgments. In this work, we present a novel method for representing the semantic content of partially written code, which can be used to guide search to solve program synthesis tasks.

Consider a tower construction domain in which a hand drops blocks, Tetris-style, onto a vertical 2D scene (Figure 1). In this domain, a function `buildColumn(n)` stacks $n$ vertically-oriented blocks at the current cursor location, and `moveHand(n)` moves the cursor $n$ spaces to the right. Given an image $\mathcal{X}$ of a scene, our task is to write a program which builds a tower matching the image $\mathcal{X}$. To do this, a model can perform search in the space of programs, iteratively adding code until the program is complete. While attempting to synthesize a program, imagine arriving at a partially-constructed program $s$ (short for *sketch*), where `HOLE` signifies unfinished code:

$$s = \texttt{loop(4, [buildColumn(1), moveHand(<HOLE>)])}$$

Note that this partial program cannot reach the goal state, because the target image has columns of height 2, but this program can only build columns of height 1. For an algorithm to determine if it should expand $s$ or explore another part of the search space, it needs to determine whether $s$ is

---

[*]Correspondence to `mnye@mit.edu`.

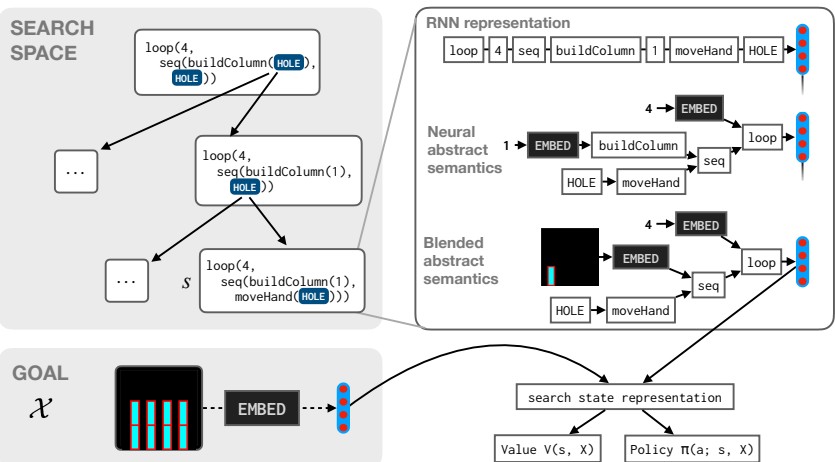

Figure 1: Schematic overview of the search procedure and representational scheme. We characterize program synthesis as a goal-conditioned search through the space of partial programs (left), and propose a novel representational scheme (blended abstract semantics) to facilitate this search process. Left: a particular trajectory through the space of partial programs, where the goal is to find a program satisfying the target image. Right: three encoding schemes for partial programs, which can each be used as the basis of a code-writing search policy and code-assessing value function.

on track to satisfy the goal. Answering this question requires an effective *representation* of partial programs.

Existing neural program synthesis techniques differ in how they represent programs. Some represent programs by their syntax (Devlin et al., 2017; Allamanis et al., 2018), producing vector representations of program structure using sequence or graph neural networks. Recently, approaches which instead represent partial programs via their *semantic* state have been shown to be particularly effective. In these **execution-guided neural synthesis** approaches (Chen et al., 2018; Ellis et al., 2019; Zohar & Wolf, 2018), partial programs are *executed* and represented with their return values. (To see why this is helpful, consider two distinct syntactic expressions $2 + 1$ and $6/2$; a syntax-based model might assign them different representations, whereas a model using a semantic representation will represent both as equivalent to $3$.) However, execution is not always possible for a partial program. In our running example, before the HOLE is filled with an integer value, we cannot meaningfully execute the partially-written loop in $s$. This is a common problem for languages containing higher-order functions and control flow, where execution of partially written code is often ill-defined.[1] Thus, a key question is: How might we represent the semantics of unfinished code?

A classic method for representing program state, known as abstract interpretation (Cousot & Cousot, 1977), can be used to reason about the set of states that a partial program could reach, given the possible instantiations of the unfinished parts of the program. Using abstract interpretation, an approximate execution model can determine if an unfinished program will eventually satisfy a goal specification. For example, in the tower-building domain, an abstract interpreter could be designed to track, for every horizontal location, the minimum tower height that all continuations are guaranteed to exceeded. However, this technique is often low-precision: hand-designed abstract execution models greatly overapproximate the set of possible execution states, and do not automatically adapt themselves to the strengths or weaknesses of specific search algorithms.

We hypothesize that, by mimicking the compositional structure of abstract interpretation, learned components can be used to effectively represent ambiguous program state. In this work, we make two contributions: we introduce **neural abstract semantics**, in which a compositional, approximate execution model is used to represent partially written code. This approach can be extended to **blended abstract semantics**, which aims to represent the state of unfinished programs as faithfully as possible by concretely executing program components whenever possible, and otherwise, approximating program state with a learned abstract execution model.

---

[1] See Peleg et al. (2020) for a discussion in the context of bottom-up synthesis.

Consider again the partial program $s$ and the blended abstract semantics encoding in Figure 1. The sub-expression `buildColumn(1)` is fully concrete, and can thus be concretely executed to render an image. On the other hand, for functions whose arguments are not fully defined, such as `moveHand`, we instead employ abstract neural modules to represent the execution state. For this example, blended neural execution makes it easy to recognize that $s$ is not a suitable partial program, because no integer argument to `moveHand`—which controls the spacing between the columns— would make the state in $s$ match the goal $\mathcal{X}$.

This combination of learned execution and concrete execution allows robust representation of partial programs, which can be used for downstream synthesis tasks. Our approach can effectively learn to represent partial program states for languages where previous execution-guided synthesis techniques are not applicable. In summary,

- We introduce blended neural semantics, a novel method for representing the semantic state of partially written programs inspired by abstract interpretation.
- We describe how to integrate our program representations into existing approaches for learning search policies and search heuristics.
- We validate our new approach with program synthesis experiments in three domains: tower construction, list processing, and string editing. We show that our approach outperforms neural synthesis baselines, solving at least 5% more programs in each domain.

## 2 RELATED WORK

Synthesizing programs from examples is a classic AI problem (Backus et al., 1957) which has seen advances from the Programming Languages community (Gulwani et al., 2017; Gottschlich et al., 2018; Solar-Lezama, 2008).

**Neurally-guided search** Recently, much progress has been made using neural methods to aid search. Enumerative approaches (Balog et al., 2016; Shi et al., 2020) use neural methods to guide an enumerative synthesizer, and can be quite suitable for small-scale domains, but can scale poorly to larger programs and domains. Translation-based techniques (Devlin et al., 2017) treat program synthesis as a sequence-to-sequence problem, and employ state-of-the-art neural sequence modeling techniques, such as recurrent neural networks (RNNs) with attention. Hybrid approaches which use sketches (Murali et al., 2017; Nye et al., 2019; Dong & Lapata, 2018) trade off computation between translation and enumeration components. These techniques can exhibit better generalization than translation-based approaches but more precise predictions than enumerative approaches (Nye et al., 2019). To combine neural learning and search, our approach follows the framework laid out in Ellis et al. (2019), where neural networks are used to guide a search over the space of possible partial programs defining a Markov decision process (MDP).

**Program representation** Prior work has studied neural representation of programs. Odena & Sutton (2019) propose property signatures to represent input-output examples, and use property signatures to guide an enumerative search. Graph neural networks have also been used to encode the syntax of programs (Allamanis et al., 2018; Brockschmidt et al., 2018; Dinella et al., 2019) for bug fixing, variable naming, and synthesis. This work has mostly focused on performing small edits to programs from real datasets. Our objective is to synthesize entire programs from specifications.

**Execution-guided synthesis** Recent work has introduced the notion of "execution-guided neural program synthesis" (Ellis et al., 2019; Chen et al., 2018; Zohar & Wolf, 2018). In this framework, the neural representations used for search are conditioned on the executed program state instead of the program syntax. These techniques have been shown to solve difficult search problems outside the scope of enumerate or syntax-based neural synthesis alone. However, such execution-guided approaches have several limitations. We aim to generalize execution guided synthesis, so that it can be applicable to a wider range of domains, search techniques, and programming language constructs.

**Abstract Interpretation** Our work is directly inspired by abstract interpretation-based synthesis (Singh & Solar-Lezama, 2011; Wang et al., 2017; Hu et al., 2020). These approaches use abstract interpretation (Cousot & Cousot, 1977) to determine if a candidate partial program is realizable under the given specification, thereby pruning the search space of programs. We see our approach as a learning-based extension to this line of work.

**Neural Modules** We employ neural module networks (Andreas et al., 2016; Johnson et al., 2017) to implement blended abstract semantics, which aims to provide a learned execution scheme inspired by abstract interpretation. This approach is also related to other tree-structured encoders (Socher et al., 2011; Dyer et al., 2016).

## 3 BLENDED ABSTRACT SEMANTICS

Consider the problem of synthesizing arithmetic expressions from input–output pairs. Suppose we have the following context-free grammar for expressions:

$$\mathcal{G} = \texttt{E} \to \texttt{E * E | E + E | x | 1 | 2 | 3 | 4}$$

and a specification $\mathcal{X}$ consisting of the input–output pairs $\{(x = 3, y = 7), (x = 5, y = 11)\}$. Suppose further that we have a candidate program $\texttt{(2 * x) + 1} \in \mathcal{G}$. To check that this program is consistent with the specification, we can evaluate it on the inputs $x$ in the specification according to the **concrete semantics** of the language: to evaluate $\texttt{(2 * x) + 1}$ on the example $(x = 3, y = 7)$, we observe that the expression $\texttt{(2 * x)}$ evaluates to the integer 6, and the expression $\texttt{1}$ evaluates to the integer 1; thus the whole expression evaluates to 7, as desired. Repeating this process with $x = 5$ returns the value 11.

Formally: Let $\mathcal{C}$ denote a context (e.g. $\{x = 3\}$)). The **concrete value** of an expression $E$ in a context $\mathcal{C}$ is:[2]

$$[\![E]\!]_\mathcal{C} = \begin{cases} [\![k]\!]_\mathcal{C} = k & \text{a constant } k \\ [\![x]\!]_{\mathcal{C} \models x = v} = v & \text{a variable } x \text{ is evaluated to its value in the context} \\ [\![f(E_1 \cdots)]\!]_\mathcal{C} = \text{run}[f, \mathcal{C}]([\![E_1]\!]_\mathcal{C} \cdots) & \text{recursively evaluate the arguments, then run } f \end{cases}$$

The goal of synthesis is to find an expression $E : [\![E]\!]_x = y$ under concrete semantics.

**Iterative construction of partial programs** Where did the expression $\texttt{(2 * x) + 1}$ come from? Neurally-guided synthesis techniques generally employ discrete search procedures. In this work, we use top-down search: starting with the top-level (incomplete) expression HOLE, we consider all possible expansions, $(\texttt{HOLE} \to \texttt{HOLE + HOLE}, \texttt{HOLE} \to \texttt{1}, \texttt{HOLE} \to \texttt{2}, \dots)$ and select the one we believe is most likely to succeed (Figure 1 left). Concrete semantics cannot be used for this selection, because expressions such as $\texttt{x + HOLE}$ are incomplete and cannot be executed. Thus, we need a different mechanism to guide search. The more effectively we can filter the set of incomplete candidate programs, the faster our synthesis algorithm will be.

Conventional **abstract interpretation** solves this problem by defining an alternative semantics for which even incomplete expressions can be evaluated. Consider the candidate expression $\texttt{HOLE * 2}$. No matter how the HOLE is filled, the expression returns an even number, so it cannot be consistent with the specification above. In many problems, we can define a space of "abstract values" (like *even integer*) and abstract semantics so that the abstract value of a partial program can be determined. This allows us to rule out partial programs on the basis of the abstraction alone (Wang et al., 2017). However, constructing appropriate abstractions is difficult and requires domain-specific engineering; an ideal procedure would automatically discover an effective space of abstract interpretations.

**Neural abstract semantics** $[\![\cdot]\!]^{nn}$ As a first step, we implement the abstract interpretation procedure with a neural network. This is a natural choice: neural networks excel at representation learning, and the goal of abstract interpretation is to encode an informative representation of the set of values that could be returned by a partial program. For the program $\texttt{1 + HOLE}$, we can encode the expression $\texttt{1}$ to a learned representation (Figure 2a, top), likewise encode HOLE (Figure 2c), and finally employ a learned abstract implementation of the + operation (Figure 2b).

For concrete leaf nodes, such as constants or variables bound to constants, neural semantics are given using a **state embedding function** EMBED$(\cdot)$, which maps any concrete state in the programming language into a vector representation: EMBED $: (State \mid \mathbb{R}^d) \to \mathbb{R}^d$. If the input to EMBED is already vector-valued, EMBED performs the identity operation. **Neural placeholders** provide

---

[2]Domains with lambdas have slightly more complicated semantics. See Appendix A for details.

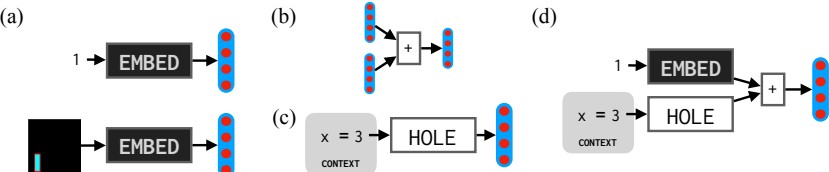

Figure 2: (a) Example applications of the EMBED function. (b) Neural abstract module for +. (c) Neural placeholder module encoding a HOLE with the context $\{x = 3\}$. (d) Neural abstract semantic encoding of the partial program $1\ +\ $ HOLE with the context $\{x = 3\}$.

a method for computing a vector representation of unwritten code, denoted by the HOLE token. To compute the representation for HOLE, we define a neural embedding function $h$ which takes a context $\mathcal{C}$ and outputs a vector. For each built-in function $f$ (including higher-order functions), the neural abstract semantics of $f$ are given by a separate **neural module** (a learned vector-valued function as in Andreas et al. (2016)) $[\![f]\!]^{nn}$ with the same arity as $f$. Therefore, computing the neural semantics means applying the neural function $[\![f]\!]^{nn}$ to its arguments, which returns a vector. Since the neural semantics mirrors the concrete semantics, its implementation does not require changes to the underlying programming language. Formally, neural semantics involve a slightly larger set of cases than concrete semantics:

$$[\![E]\!]_{\mathcal{C}}^{nn} = \begin{cases} [\![k]\!]_{\mathcal{C}}^{nn} = \text{EMBED}(k) & \text{a constant } k \text{ is embedded} \\ [\![x]\!]_{\mathcal{C}\models x=v}^{nn} = \text{EMBED}(v) & \text{embed the value } v \text{ of the variable } x \\ [\![\texttt{HOLE}]\!]_{\mathcal{C}}^{nn} = h(\mathcal{C}) & \text{a neural placeholder based on context} \\ [\![f(E_1 \cdots)]\!]_{\mathcal{C}}^{nn} = [\![f]\!]^{nn}([\![E_1]\!]_{\mathcal{C}}^{nn} \cdots) & \text{using neural module } [\![f]\!]^{nn} \end{cases}$$

This encoding is only one way to define a neural semantics, adopting a relatively simple and generic representation for all program components. For a discussion of its limitations and other, more sophisticated representations that could be explored in future work, see Appendix C.

**Blended abstract semantics** $[\![\cdot]\!]^{blend}$   Notice that for an expression such as (2 * x) + HOLE, the concrete value of the sub-expression (2 * x) is known, since it contains no holes. The neural semantics above don't make use of this knowledge. To improve upon this, we extend neural semantics and introduce blended semantics, which alternates between neural and concrete interpretation as appropriate for a given expression:

- If the expression is a constant or a variable, use the concrete semantics.

- If the expression is a HOLE, use the neural semantics.

- If the expression is a function call, recursively evaluate the expressions that are the arguments to the function. If all arguments evaluate to concrete values, execute the function concretely. If any argument evaluates to a vector representation, transform all concrete values to vectors using EMBED and apply the neural semantics of the function.

Formally, we can write:

$$[\![E]\!]_{\mathcal{C}}^{blend} = \begin{cases} [\![k]\!]_{\mathcal{C}}^{blend} = [\![k]\!]_{\mathcal{C}} = k & \text{a constant } k \\ [\![x]\!]_{\mathcal{C}\models x=v}^{blend} = [\![x]\!]_{\mathcal{C}\models x=v} = v & \text{variable } x \text{ in context} \\ [\![\texttt{HOLE}]\!]_{\mathcal{C}}^{blend} = [\![\texttt{HOLE}]\!]_{\mathcal{C}}^{nn} = h(\mathcal{C}) & \text{a neural placeholder based on context} \\ [\![f(E_1 \cdots)]\!]_{\mathcal{C}}^{blend} = [\![f]\!]([\![E_1]\!]_{\mathcal{C}}^{blend} \cdots) & \text{if all arguments are concrete} \\ [\![f(E_1 \cdots)]\!]_{\mathcal{C}}^{blend} = [\![f]\!]^{nn}(\text{EMBED}([\![E_1]\!]_{\mathcal{C}}^{blend}) \cdots) & \text{if any arguments are vectors} \end{cases}$$

Because blended abstract semantics replaces concrete sub-components with their concrete values, we expect blended semantics to result in more robust representations, especially for long or complex programs where large portions can be concretely executed.

## 4    PROGRAM SYNTHESIS WITH BLENDED ABSTRACT SEMANTICS

To perform synthesis, we experiment with methods to guide search introduced in Ellis et al. (2019).[3] In this work, the search over partial programs is formulated as an MDP, in which each state is a pair $(s, \mathcal{X})$ consisting of a partial-program and a specification, and actions $a \in \mathcal{G}$ are expansions of HOLEs under rules under the grammar. We assume a reward of 1 for programs which satisfy $\mathcal{X}$. In this framework, we **learn to search** by training a policy $\pi(a|s, \mathcal{X})$ that proposes expansions to $s$, and optionally a value function $V(s, \mathcal{X})$ that predicts the probability that $\mathcal{X}$ is solvable via any expansion of $s$.

Let $\mathcal{X} = \{(x_i, y_i)\}$, where $(x_i, y_i)$ are input-output pairs. Let $[\![s]\!]_{x_i}^{blend}$ denote the blended abstract semantic representation of $s$ with input $x_i$. The representation of a state $\texttt{rep}(s, \mathcal{X})$ is:

$$\texttt{rep}(s, \mathcal{X}) = \frac{1}{n} \sum_{x_i, y_i \in \mathcal{X}} \texttt{ReLU}(W([\![s]\!]_{x_i}^{blend}; \textsc{Embed}(y_i)]))$$

Here, $W$ is a learnable weight matrix, and the representation is averaged across all input-output pairs of $\mathcal{X}$. Given this state representation, the policy and value function are:

$$\pi(a \mid s, \mathcal{X}) = \texttt{softmax}(\texttt{MLP\_a}(\texttt{rep}(s, \mathcal{X})))$$
$$V(s, \mathcal{X}) = \sigma(\texttt{MLP\_V}(\texttt{rep}(s, \mathcal{X})))$$

Here, MLP is a multi-layer perceptron. Note the value function outputs a value between 0 and 1; this allows for a probabilistic interpretation (see below).

**End-to-end training**    We train our policy $\pi$ using imitation learning. Starting from the empty partial program $s_0 = \texttt{HOLE}$, we generate a sequence of partial programs $s_1, s_2, \cdots$ by sampling a sequence of expansions $a_0, a_1, \cdots$ [4] from the grammar $\mathcal{G}$. Let $p = s_T$ be the completed program. We obtain specifications $\mathcal{X} = \{(x_i, y_i)\}$ by sampling a set of inputs $x_1 \cdots x_n$ and obtaining outputs using concrete semantics $y_i = [\![p]\!]_{x_i}^c$. Thus, from a sequence of expansions $a_0, a_1, \cdots$, we can collect a set of triplets $\{(\mathcal{X}, s_i, a_i)\}$ as training data. This process is repeated to generate the training set $\mathcal{D}$. We can then perform supervised training, maximizing the log likelihood of the following:

$$\mathcal{L}(\pi) = \mathbb{E}_{(\mathcal{X}, s, a) \sim \mathcal{D}} \left[ \log \pi(a \mid s, \mathcal{X}) \right]$$

We train the value function by sampling rollouts of partial programs $s_0 \cdots s_T$ from a fully-trained $\pi$, minimizing the error in a Monte-Carlo estimate of the expected reward $R$ (i.e., the probability of success under the policy).

$$\mathcal{L}^{\text{RL}}(V) = \mathbb{E}_{(R, s_0 \ldots s_T) \sim \pi(\cdot | s_0, \mathcal{X})} \left[ \sum_{t \leq T} \text{err}(V(s_t, \mathcal{X}), R) \right]$$

For our error function, we use logistic loss rather than the more common mean squared error (MSE).

**Search**    We explore a variety of code-writing search algorithms. Using only a policy, we can employ sample-based search and best-first search (where the log probability of generating $s$ under $\pi$ is used as the scoring function). With the addition of a learned value function, we can perform A\*-based search with $-\log V(s, \mathcal{X})$ as a heuristic (see Ellis et al. (2019) for details).

## 5    EXPERIMENTS

We evaluate our model in two domains containing language constructs not handled by concrete execution-guided synthesis approaches: a tower-building domain with looping constructs, and a list-processing domain with higher order functions. We additionally test on a string-editing domain for which execution-guided synthesis is possible, but requires language modification; there, we examine how our approach fares without these modifications.

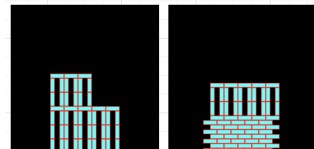

Figure 3:    Example tower-building constructions.

---

[3]We briefly review these methods here, but see Ellis et al. (2019) for more details.

[4]Often a PCFG is used to sample these expansions. See Appendix B for details.

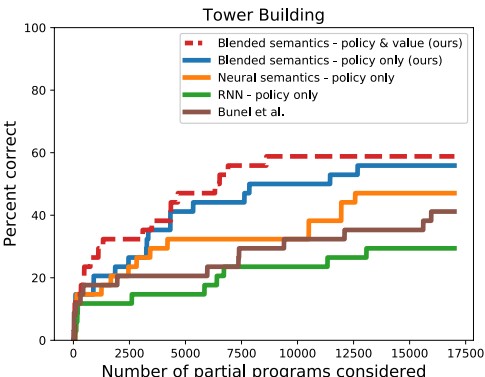 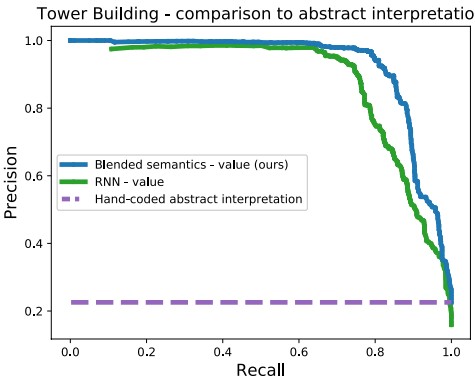

Figure 4: Left: Overall synthesis results in the tower-building domain. We plot the percentage of test problems solved as a function of the number of partial programs considered per synthesis problem. Right: Comparing the value function to a hand-coded abstract interpretation. Blended abstract semantics outperforms baselines in synthesis tasks, and obtains higher classification precision than hand-coded abstract interpretation.

## 5.1 LOOPING CONSTRUCTS: TOWER CONSTRUCTION

We begin by investigating how our model performs in generative programming domains with higher-level control flow such as loops. Looping programs are an essential part of sophisticated programming languages, and aren't naturally handled by previous execution-guided synthesis approaches. Our experiments in the tower-building domain employ a domain specific language (DSL) similar to the language depicted in the introduction to construct towers in a 2D world, adapted from Ellis et al. (2020). This domain is inspired by classic AI tasks (Winston, 1972). It is also related to important problems in the AI literature, such as generalized planning, where plans can be represented as programs, and often require looping constructs (Jiménez et al., 2019; Srivastava et al., 2015).

As above, the goal is to construct a program which successfully renders to a target image (examples in Figure 3). Language details can be found in Appendix B. We compared against two ablation models (see Figure 1): (1) Neural abstract semantics (defined in Section 3), which does not apply concrete execution to concrete subtrees, and (2) RNN encoding, which encodes partial programs using a gated recurrent unit (GRU): $\pi(a \mid \texttt{GRU\_enc}(s), \mathcal{X})$. We also consider an additional encoder-decoder baseline. This baseline uses a convolutional neural network (CNN) to encode the specification image, and then employs an LSTM decoder to decode the tokens of the target program, while attending over the image representation via Spatial Transformer Network attention (Jaderberg et al., 2015). This baseline is inspired by the architecture used for the Karel domain in Bunel et al. (2018), with the addition of spatial attention. We train all models on 480,000 programs sampled from the DSL. More details can be found in Appendix B.

To evaluate our model, we constructed a test set of tower-building problems involving combinations of tower-building motifs seen during training. Tower objects seen during training were composed in previously unseen ways, by stacking towers on top of each other, or placing them side-by-side. We evaluate our models by performing best-first search from the learned policy. We also evaluate a *value-guided* heuristic search using the A* algorithm with the negative log likelihood of the policy as the prior cost and the value function as the future cost estimate.

**Synthesis results** Figure 4 (left) shows our overall synthesis results in the tower-building domain, measuring the percentage of test problems solved as a function of the number of search nodes (partial programs) considered per problem. The sequence encoding performs poorly and is unable to solve a majority of test problems. The neural abstract semantics model achieves better performance, solving about half of the test problems within the allotted search budget. Blended execution outperforms each baseline. We also observe that adding a value function as a search heuristic further increases performance of our blended model, which is consistent with the findings in Ellis et al. (2019).

**Comparison to abstract interpretation** How does the learned value function compare to hand-coded abstract interpretation? During synthesis, we can use abstract interpretation to prune the search space by rejecting candidate partial program candidates for which the desired output state is not within the abstract state achieved by executing the partial program (Singh & Solar-Lezama, 2011; Wang et al., 2017; Hu et al., 2020). Used in this way, classic abstract interpretation is conservative; it can be thought of as a classifier with perfect recall, but poor precision, only rejecting the partial programs it knows for sure to be unsuitable. Can our value function also detect these clearly bad partial programs, but ascribe low value to less obviously bad candidates? To test this, we conditioned the model on tasks from our test corpus, and sampled 15 search trajectories from our blended semantics policy for each task. For each partial program encountered during search, we compute the model's value judgment, and recorded whether each rollout was successful. Treating rollout success as a noisy label of partial program quality, and using the value function as a classifier, we plot precision vs recall of the value judgements as we vary the classification threshold. Figure 4 right shows our results for this experiment. As the classification threshold is varied, our learned value maintains comparable recall compared to the hand-coded abstraction, while achieving better precision. For high classification thresholds, our model achieves performance comparable to the hand-coded abstract interpretation, and additional precision is gained by lowering the classification threshold. The RNN value performs worse on this test, achieving lower precision and recall.

## 5.2 HIGHER-ORDER FUNCTIONS: FUNCTIONAL LIST PROCESSING

| List Processing | | String Editing | |
| --- | --- | --- | --- |
| Examples | Program | Examples | Program |
| `[-4,9,4,6] → [18,12]`
`[15,3,3,-14] → [30,6,6]` | `map (λx.x*2) (filter`
`  (λx.(x>0 & x%3==0)) input)` | `+106 769-858-438 → (769)`
`+63 099-824-351 → (099)` | `Const('(') \| GetToken(Number, 1)`
`  \| Const(')')` |
| `[1,2,3,4] → [5,5,5,5]`
`[1,2,4,-1] → [0,6,6,0]` | `zipwith (λx,y.x+y)`
`  (input) (reverse input)` | `Mariya Sergienko → Dr. Mariya`
`Andrew Cencici → Dr. Andrew` | `Const('D') \| Const('r')`
`  \| Const('.') \| Const(' ')`
`  \| GetToken(Word, 0)` |

Figure 5: Example programs from the list processing (left) and string editing (right) domains.

In our second experimental domain, we seek to answer two questions: How well does our model perform on input-output synthesis? How effectively can it synthesize programs containing higher-order functions? Although previous work (Zohar & Wolf, 2018) has successfully applied execution-guided approaches to list processing (using the DeepCoder language), the use of higher-order functions was severely limited: only a small, predefined set of "lambdas," (such as `(*2)`, `is_even`, `(>0)`) were used as arguments for higher-order functions. For example, synthesizing a program which "filters all elements divisible by 3 from a list" is not possible with this DSL. However, in real programming languages, higher-order functions must be able to accept a combinatorially large set of possible lambda functions as input. This presents a challenge for execution-guided synthesis approaches such as

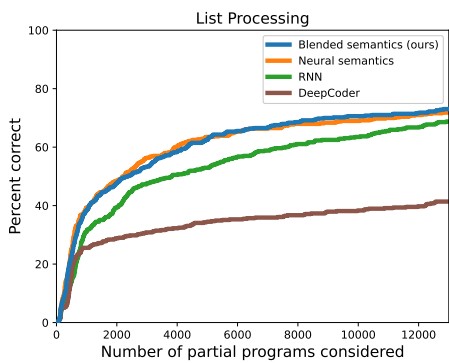

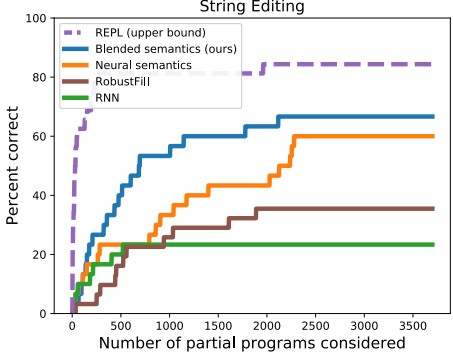

Figure 6: Synthesis results for list processing. Models were trained and tested on programs each containing 2-3 higher order functions with lambdas of depth 3.

Figure 7: Synthesis results for string editing. Blended semantics outperforms all baselines, except for the execution-guided REPL model, which relies on domain-specific language modifications.

Zohar & Wolf (2018), for which the assumption of a small set of lambda functions is key. To this end, we modified the Deepcoder DSL to allow a richer set of possible programs. We replaced the predefined set of lambda functions with a grammar allowing for the combinatorial combination of grammar elements (examples in Figure 5 left). The modified grammar is given in Appendix B. All models were trained on 500,000 training programs.

**Results**    Figure 6 shows the results of synthesis using best-first search from a policy on test problems sampled the same distribution as the training problems. Our blended model and our neural semantics model both outperform the RNN baseline, achieving 5-10% higher accuracy given the same search budget. We additionally implemented the DeepCoder model (Balog et al., 2016), which conditions the search only on input-output examples and not partially constructed programs. This model achieves considerably lower accuracy for a given search budget. The blended model also yielded superior results on numerous variations of these tasks (increasing number of higher order functions, varying integer ranges, performing sample-based search, etc). In the sample-based search condition, we also compare against a RobustFill baseline, which is outperformed by blended semantics. See Figure 8 in Appendix B for details.

## 5.3    IO PROGRAMMING: STRING EDITING

In our final experiment, we examine how our model performs on domains for which execution-guided synthesis is possible, but requires extensive changes to the underlying DSL.

For example, in the RobustFill DSL, a function `getSubStr(i,j)` slices a string from index $i$ to index $j$. This function is not executable until both $i$ and $j$ are known. In order to perform execution-guided synthesis, Ellis et al. (2019) needed to replace `getSubString` with two separate functions: `getSubStrStart_i` and `getSubStrEnd_j`, where each half can be executed in the read-eval-print loop (REPL). This process must be performed manually for every language construct which takes multiple arguments.

Here we seek to determine whether our model can synthesize programs using the language as-is. To this end, we implement the code-writing policy using the DSL presented in Devlin et al. (2017) without modification (example programs in Figure 5 right). Because the REPL system in Ellis et al. (2019) uses a domain-specific, hand-designed partial program executor in-the-loop, we do not expect our approach (which must learn to approximate the semantics) to outperform the REPL system; however, it can recover some of the REPL system's improvement over ordinary (syntax-based) synthesis. We additionally compare against another relevant baseline: RobustFill (Devlin et al., 2017). In contrast to the original paper, we train the RobustFill model using the same "unmodified" version of the DSL as our model, whose syntax has not been modified to aid with prediction. We train all models on 2 million programs sampled from the DSL. At test time, we used a sample-based search procedure, because the branching factor is prohibitively large for breadth-first search procedures explored above. While the blended encoding does not achieve the accuracy of the execution-guided REPL system, it outperforms the other baselines, including the RobustFill model, neural abstract semantics and the RNN baseline.

## 6    CONCLUSION

We introduced blended abstract semantics, a method for representing partially written code based on concrete execution and learned approximate semantics. We demonstrated how our approach, which combines abstract interpretation with representation learning, can be trained end-to-end as the basis for search policies and search heuristics. In program synthesis tasks, models equipped with blended abstract semantics outperformed neural baselines in several domains. Immediate future directions include exploring the use of blended abstract semantics for other synthesis tasks, including programming from language instruction and bug fixing. More generally, we hope that approaches which integrate learning and symbolic methods can be used to build systems which can write code more effectively and robustly.

ACKNOWLEDGMENTS

The authors gratefully acknowledge Kevin Ellis and Eric Lu for productive conversations. We additionally thank anonymous reviewers for helpful comments. M. Nye is supported by an NSF Graduate Fellowship and an MIT BCS Hilibrand Graduate Fellowship.

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

## A    SEMANTICS

Here we fully define the semantics for concrete, neural, and blended semantics, covering details that were omitted in the main paper, namely, the semantics of lambda expressions.

**Concrete semantics**    In this work, we consider domains where the underlying programming language is *functional*. Let $\lambda x.E$ be a lambda expression of one argument, and let $x = v$ be an assignment of the variable $x$ to the value $v$. Lambda application is defined as follows:

$$(\lambda x.E)(v) = [\![E]\!]_{\{x=v\}}$$

That is, we evaluate the function body $E$, replacing all instances of the function variable $x$ with the value $v$. For example, $[\![(\lambda x.x + x)(5)]\!]_{\{\}} = [\![x + x]\!]_{\{x=5\}} = 5\hat{+}5 = 10$, where $\hat{+}$ denotes the execution semantics of the built-in function $+$. The concrete semantics $[\![\cdot]\!]$ is defined:

$$[\![E]\!]_{\mathcal{C}} = \begin{cases} [\![k]\!]_{\mathcal{C}} = k & \text{a constant } k \\ [\![x]\!]_{\mathcal{C} \models x=v} = v & x = v \\ [\![f(E_1 \cdots)]\!]_{\mathcal{C}} = \hat{f}([\![E_1]\!]_{\mathcal{C}} \cdots) & \text{executing built-in } f \\ [\![(\lambda x_1 \cdots x_n.E)(E_1, \cdots, E_n)]\!]_{\mathcal{C}} = [\![E]\!]_{\mathcal{C} \cup \{x_1=[\![E_1]\!]_{\mathcal{C}}, \cdots, x_n=[\![E_n]\!]_{\mathcal{C}}\}} & \text{lambda application} \end{cases}$$

**Neural semantics**    For built-in functions $f$, we use $f^{nn}$, a neural module function of $k$ vector inputs, where $k$ is the arity of $f$. We note that our treatment of functions discussed here applies to both first-order and higher-order functions.

In our domains, lambda expressions are only used as arguments to higher-order functions. Therefore, since we will never apply a lambda expression directly under neural semantics, we only require vector representation of lambdas. However, we still require a mechanism to represent arbitrary lambda expressions built combinatorially from primitive functions. This representation is constructed in a modular fashion by encoding the body $E$ of the lambda expression.

$$[\![E]\!]_{\mathcal{C}}^{nn} = \begin{cases} [\![k]\!]_{\mathcal{C}}^{nn} = \text{EMBED}(k) & \text{a constant } k \text{ is embedded} \\ [\![x]\!]_{\mathcal{C} \models x=\text{null}}^{nn} = \text{EMBED}(\text{``}x\text{''}) & \text{embed the variable } x \text{ when the value is null} \\ [\![x]\!]_{\mathcal{C} \models x=v}^{nn} = \text{EMBED}(v) & \text{embed the value } v \text{ of the variable } x \\ [\![\text{HOLE}]\!]_{\mathcal{C}}^{nn} = h(\mathcal{C}) & \text{a neural placeholder for a hole based on context} \\ [\![f(E_1 \cdots)]\!]_{\mathcal{C}}^{nn} = f^{nn}([\![E_1]\!]_{\mathcal{C}}^{nn} \cdots) & \text{using neural module} \\ [\![\lambda x.E]\!]_{\mathcal{C}}^{nn} = [\![E]\!]_{\mathcal{C} \cup \{x=\text{null}\}}^{nn} & \text{encode the body of a lambda} \end{cases}$$

Note that in representing a lambda expression, we used the context with the assignment $x = $ null. This is used to account for the fact that, at the time of the lambda function's definition (as an argument to a higher-order function), its argument is still unknown under neural execution. Likewise, to represent a variable whose value has been assigned to null, we return a vector representing the variable.

The definition for blended semantics proceeds in an analogous fashion, with concrete subtrees executed concretely.

## B    EXPERIMENTAL DETAILS

All models are trained with the AMSGrad (Reddi et al., 2018) variant of the Adam optimizer with a learning rate of 0.001. All RNNs are 1-layer and bidirectional GRUs, where the final hidden state is used as the output representation. Unless otherwise stated, holes are encoded by applying the EMBED function to the context, and then applying a type-specific neural module to the resulting vector. For the tower and string-editing domains, which use continuation-passing style, holes are filled in left-to-right. For the list editing domain, holes are filled in right-to-left.

### B.1    TOWERS

We employ the tower-building domain and DSL introduced in Ellis et al. (2020), which consists of the basic commands: `PlaceHorizontalBlock`, `PlaceVerticalBlock`, `MoveHand`,

`ReverseHand`, `Embed`, `Loop` and integers `n` from 1 to 8. (The higher-order `Embed` function takes an expression as input, executes it, and then returns the hand to its initial location.)

Formally, the syntax of the tower-building domain is given as follows:

```
P = [S]
S = embed(P) | moveHand(n) | reverseHand() | loop(n, P) |
placeHorizontalBlock() | placeVerticalBlock()
n = 1..8
```

A program `P` is executed sequentially one statement (`S`) at a time, with each statement modifying the state of the tower construction. In a tower construction, a state is defined as follows:

```
state:  (handLoc, handOrientation, history)
history:  [(loc, blockType)]
Loc:  (x:int, y:int)
```

We assume the initial tower state is: `s = (0, 1, [])`

The semantics of applying each statement S applied to a state `s` is as follows:

```
placeHorizontalBlock = placeBlock(3, 1)

placeHorizontalBlock = placeBlock(1, 3)

placeBlock(w:int, h:int) : s:state ->
  (s.handLoc, s.handOrientation, s.history +
  ((s.handLoc.x,
  topY(s.history, s.handLoc.x, s.handLoc.x+w) ), w, h))

topY(history, xlow, xhigh) : max( y |  y = y'+h
  where ((x',y'), w,h) in history
  and [x, x+w] intersect [xlow, xhigh])

moveHand(dist:int): s:state ->
  (s.handLoc + s.handOrientation*dist, s.handOrientation, s.history)

reverseHand(): s:state ->
  (s.handLock, s.handOrientation*-1, s.history)

loop(n:int, fn): s:state -> fn^n(s)

embed(fn): s:state ->
  let s' = fn(s) in (s.handLoc, s.handOrientation, s'.history)
```

In order to test the compatibility of our approach with library-learning techniques, we additionally use library functions learned by the DreamCoder system by combining the above functions. Following Ellis et al. (2020), the grammar is implemented in continuation-passing style. Our training data consisted of tower programs randomly sampled from a PCFG generative model (Ellis et al., 2020).

We trained policy networks on 480000 programs. We trained value functions on 240000 rollouts from the policy. We perform search for up to 300 seconds per problem.

For the blended semantics and neural semantics models, all neural modules consist of a single linear layer (input dimension $512 * n_{args}$ and output dimension $512$) followed by ReLU activation. Tower images are embedded with a simple CNN-ReLU-MaxPool architecture, as in Ellis et al. (2020).

For the tower-building domain, we also include a baseline inspired by Bunel et al. (2018). This model uses a CNN image encoder to encode the spec tower image. The image representation is passed through an MLP layer and initializes an LSTM decoder, which sequentially decodes the program tokens. As an additional modification to the original model from Bunel et al. (2018), at each decoding step we perform spatial attention over the image encoding via a Spatial Transformer Network (Jaderberg et al., 2015). Our CNN image encoder consists of 4 convolutional layers (Conv

+ ReLU) with kernel size 3, and a single max-pool of size 2 after the first convolution. Our LSTM decoder has hidden size 512 and embedding size 128.

**Hand-coded abstract interpretation**   We implemented an abstract domain which tracked, a) the range of possible locations of the "hand" and b) for each horizontal location, the minimum height which must be achieved by the partially constructed tower. This representation allows us to eliminate invalid partial programs because once a block is dropped, it cannot be removed through any subsequent commands.

## B.2   LIST PROCESSING

Data for this domain was generated by modifying the DeepCoder dataset (Balog et al., 2016). Specifically, DeepCoder training programs of size 2 (containing 2-3 higher order functions, such as `map f (filter g input)` or `zipwith f (map g input) (map h input)` ) were modified by changing the lambdas in the program (`f`, `g`, and `h` in the above examples) from a small set of constant lambdas such as (`*2`) to depth-3 lambdas sampled from our modified grammar (see below). For example: (`λx.max(x+2,x/2)`). For each program, 5 example input lists were sampled, each with length 10 and values in the range [-64, 64]. The program was then executed to yield the corresponding outputs. Programs with output or intermediate values outside of the range [-64, 64] were discarded. Programs producing the identity function or constant functions were also discarded. We trained and tested only on functions of type `[int]` $\rightarrow$ `[int]`. At test time when running search, we similarly reject programs with intermediate values outside of the desired integer range.

All policy networks were trained on 500000 programs. We perform search for 180 seconds per problem.

All neural modules consist of a single linear layer (input dimension $64 * n_{args}$ and output dimension $64$) followed by ReLU activation. Integers are encoded digit-wise via a GRU. Lists are encoded via a GRU encoding over the representations of the integers they contain.

Unbound variables within a lambda function are embedded via a learned representation parameterized by the variable name (one vector representing x and one representing y). When encoding holes within lambda functions, we ignore context, and instead embed holes only as a function of the hole type.

The DeepCoder baseline is based on Balog et al. (2016) but uses the same input-output example encoding architecture as the other methods implemented here. Our implementation uses an MLP with 1 64-dimensional hidden layer with a tanh activation to produce the production rule probabilities from the input-output encodings.

In the sample-based search condition in Figure 8, we additionally compare against a RobustFill model. Our RobustFill implementation is equivalent to the "Attention-A" model from Devlin et al. (2017). We use GRUs of hidden size 64 and embedding size 64, in keeping with the models above.

**Modified lambda grammar**   Below is the grammar used for lambda functions:
```
L → (λx,y.S) | (λx.S)
S → I | B
I → I+I | I*I | I/I | min(I,I) | max(I,I) | A
B → I > I | or(B,B) | and(B,B) | I%I==0
A → x | y | N
N → -2 | -1| 0 | 1 | 2
```

**Variations on training and testing conditions**   Many variations on the training and testing conditions achieve similar results to those shown in the main paper (i.e., blended semantics consistently achieves the highest performance). Several of these variations are shown in Figure 8.

### B.3 STRING EDITING

For the string editing tasks, we use the DSL from Devlin et al. (2017). We train on randomly sampled programs, sampling I/O pairs and propagating constraints from programs to inputs to ensure that input strings are relevant for the target program (see Devlin et al. (2017)). We condition on 4 I/O examples for each program. We used string editing problems from the SyGuS (Alur et al., 2016) program synthesis competition as our test corpus. We trained all models on 2 million training programs. At test time, we sample programs from the model for a maximum timeout of 30 seconds.

Input and output strings are encoded by embedding each character via a 20-dimensional character embedding and concatenating the resulting vectors to form a representation for each string. Representations of "expressions" $e$ (as defined in the RobustFill DSL) are concatenated together using an "append" module. Following Ellis et al. (2019), neural modules consist of a single dense block with 5 layers and a growth rate of 128 (input dimension $256 * n_{args}$ and output dimension 256).

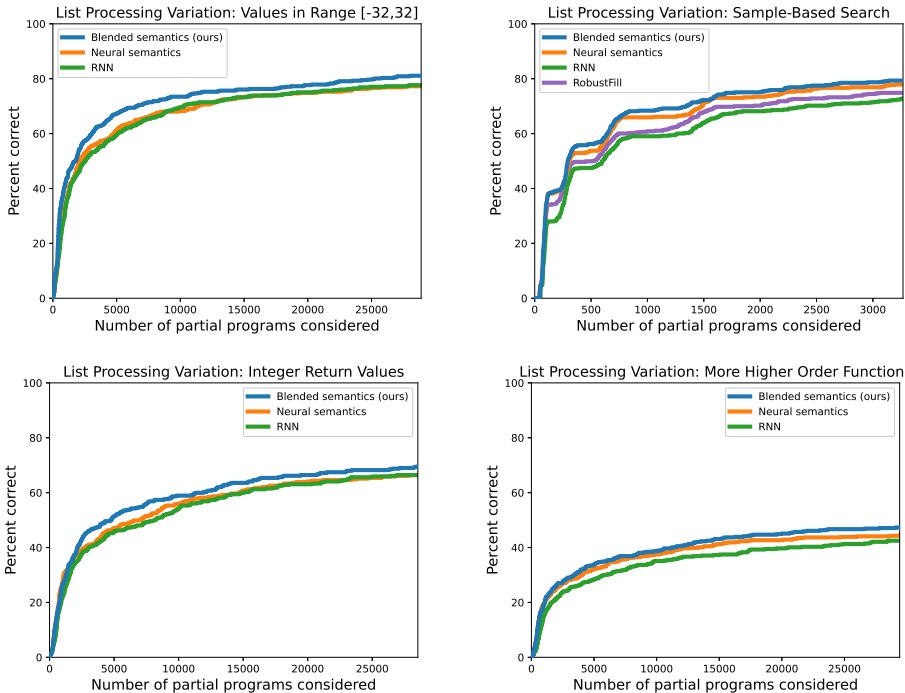

Figure 8: Variations on the list processing task. Top Left: using integer values in the range [-32, 32] instead of [-64, 64]. Top Right: using sample-based search instead of best-first search, including a comparison to RobustFill. Bottom Left: extending the training and testing data to allow for `[int]` → `int` functions and with [-32,32] as the range. Bottom Right: the original model tested on deeper DeepCoder programs with 3-6 higher-order functions.

## C    LIMITATIONS OF OUR NEURAL SEMANTICS

Our approach to neural semantics differs conceptually in several ways from abstract interpretation (Cousot & Cousot, 1977), and the way it has previously been used to constrain symbolic search for program synthesis (Singh & Solar-Lezama, 2011; Wang et al., 2017; Hu et al., 2020). Two main differences stand out as potential limitations of our approach, and avenues to be explored further in future work. Firstly, our method treats loops generically, identically to other higher-order functions. This means neural modules must compute a representation of a looping program using a fixed, feed-forward computation, without an iterative fixpoint computation. It is possible that a more rigorous treatment of loops could lead to improved accuracy, though it would introduce additional complexity to the method. We believe that this may be an interesting direction for future work. Secondly, our handling of lambda functions also differs from abstract interpretation. As discussed in Appendix A,

when using a lambda as an argument to a higher-order function, our method represents the lambda expression as a vector, without encoding the values of its arguments. A method which can treat lambdas more generically would also be an interesting direction for future work. We thank an anonymous reviewer for highlighting these two points.

