# OpenReview forum: "Representing Partial Programs with Blended Abstract Semantics"
_ICLR.cc/2021/Conference — ICLR 2021 Poster_

### Official Review · AnonReviewer4 · 2020-10-23
**Neural partial evaluation of sketched code**

**Rating:** 7
**Confidence:** 4

**Review:**

# Positives

* (+1) Blended neural semantics is an elegant and intuitive construct.
* (+2) Performance seems to be good for interesting and non-trivial DSLs.

# Negatives
* (-1) Although the high-level description of the approach is intuitive, it is lacking many details. The experimental evaluation is distressingly vague.
* (-2) Comparison to baselines is poor. Although the main approach is compared to baselines introduced here (remove blended semantics, use just an RNN), it is not compared to most prior approaches on the benchmarks of those prior approaches.

1. I like this work and the main ideas are intuitive and well described. However, I found the paper disturbingly lacking in detail. I see some of that detail in the appendix, but information such as dataset size is essential in understanding results. It can't be in the appendix only.
1. Also, the paper seems to make some off-hand value judgments (pun intended) about prior approaches, without substantiation. For example, it claims that execution-guided synthesis isn't compatible with loop constructs, but no explanation is given for this. Similarly, the paper says that it can't possibly compete with REPL on an I/O synthesis task on the RobustFill DSL, but not explanation is given. This makes for a jarring reading experience.
1. Page 1, paragraph 2. Your introduction would be much more effective if you explained briefly the semantics for your DSL (even though it's relegated to the appendix). Otherwise, your example doesn't make sense. I don't know for instance what, precisely, ` buildColumn(1)`  means. Is `1`  the height from the bottom? Or the height of a column addition? Can I stack columns? Your text, for example, seems to imply that `buildColumn(1)` has no place in a solution to the example problem, because the column has height 1 (sentence right after the sketch example). However, the goal state on Figure 1 shows stacked columns of height 1. It's even more confusing that your appendix description of the DSL implies there are learned functions (presumably including `buildColumn`), which aren't shown here. All in all, this introduction confused me more than it sold me your work.
1. Page 2, paragraph 1. Somehow this text seems to suggest that higher-order functions and control flow are incompatible with approaches that don't know how to evaluate partial programs. That's not true, of course. REPL could do just fine in such a setting (in principle) but it would have to build the program bottom up, with all of the arguments of a higher-order function, before it composed them into an application of the higher-order function. What you haven't motivated yet is *why* we need to write unfinished code, rather than bags of expressions from which to compose the finished program. You haven't given any intuition or evidence of that. I suspect that the intuition comes from the fact that not all execution models can be interpolated (along the lines of string manipulation tasks or shape composition tasks, a la REPL), and therefore you can't just look at the output and expect to get good hints of the building blocks you need to compose greedily. Sometimes you need to choose a code idiom (a loop) before you choose what goes into the loop. It would help your paper if you made an argument why that makes sense in many cases.
1. Page 4. What's the embedding function for concrete expressions?
1. Page 4. What's a precise definition of context? In the examples, you've shown value assignments to variables, but that doesn't allow for having multiple `HOLE`s in a program. Do you only allow one `HOLE`? The examples show multiple `HOLE`s, which suggests there must be some positional encoding involved, or different `HOLE`s would embed to the same vector given the same context.
1. Page 4. Are neural modules separate for every primitive operation? Even for learned functions? How many are there?
1. Page 6, Section 5.0. Why exactly aren't these domains covered by execution-guided synthesis? This is an oft stated limitation of prior approaches, but no evidence has been given. Why wouldn't REPL, for instance, fit the block world?
1. Page 6, how was test construction exactly? "combinations of motifs seen during training" isn't very precise.
1. Page 6. The policy isn't a cost function. You mean, presumably, the probability associated with the chosen action? Something else? Please explain.
1. Figure 4. Please explain your x axis. What are programs considered? In total? Per problem? After how much training? How big are the datasets?
1. Page 7. It seems a pity that you don't also compare to DeepCoder and FlashFill on the same datasets, rather than only extending the DSLs for your setting. It would be valuable to know how the approach works in those simpler DSLs.
1. Section 5.3, 3rd paragraph. The string DSL is on the right, not on the left in Figure 5.
1. Section 5.3. "We do not expect that our approach would outperform the REPL..." Why not? Explain.

---

> ### Author Response · Authors · 2020-11-19
> **Response to Reviewer 4 (1/2)**
>
> Thank you very much for your helpful comments and positive reception of our work! Our response is below.
>
> Your review requests clarification for two important questions: Why is execution-guided synthesis incompatible with higher-level control flow? Why do we expect execution-guided synthesis methods (such as REPL) to outperform other learned methods, such as blended semantics, in domains for which they are applicable? We provide answers below, and update the draft accordingly.
>
> - **Why is execution-guided synthesis incompatible with higher-level control flow?**
> Execution-guided synthesis builds programs bottom-up. It is a well-known limitation of bottom-up synthesis that when evaluating a lambda function used within higher-order functions, the lambda function cannot be executed directly, because the arguments to the lambda function are unknown (for more details, see Appendix C and Appendix D in Peleg et al., 2020: https://cseweb.ucsd.edu/~hpeleg/resl-oopsla20.pdf). Recent work has attempted to address this in some specific contexts (Peleg et al., 2020), but it remains an open problem.
> To see this clearly, we can construct the following example program (in python syntax):
> ```
> map( lambda x: x**2 + 5*x - 17, lst)
> ```
> In order to construct this program bottom-up in a piecewise manner, by first constructing the expression `lambda x: x**2 + 5*x - 17` and `lst` , and then applying `map`, we would need to be able to first execute the lambda expression. However, this is not possible without the value of the input `x`, which cannot be determined until the higher-order function `map` and argument `lst` are determined. Therefore, a method which performs execution-guided bottom-up synthesis would be required to synthesize this entire program at once. The same argument can be made for the following looping function (again, using python syntax for clarity):
> ```
> for i in range(5):
>     placeBlock()
>     moveHand(i+2)
> ```
> You can’t execute this loop without knowing the content of the body, but the body depends on the loop iteration count (`moveHand(i+2)`). We note that it is often possible to refactor a given language, so that it is compatible with execution guided synthesis (e.g., the string editing domain in Ellis et al., 2019), however this requires extensive domain-specific hand-engineering.
> - **Why do we expect execution-guided synthesis methods (such as REPL) to outperform other learned methods, such as blended semantics, in domains for which they are applicable?**
> Techniques such as blended semantics and neural semantics must learn an approximation of the semantics in order to form representations of the semantic content from the syntax. On the other hand, execution-guided synthesis employs the exact execution model in-the-loop during search. We expect the learned approximate execution models to be worse than the exact model used by execution-guided synthesis.
>
> Other questions and comments:
>
> - We have added details to clarify the semantics of the tower-building domain in Appendix B. We have also added clarifying points in the introduction to make for a smoother reading process. In particular, in our introduction, `buildColumn(n)` means to stack n vertically-oriented blocks on top of each other. Therefore, the goal state in Figure 1 could be achieved by looping over the expression `[buildColumn(2), moveHand(2)]` for 4 iterations.
>
> - **“Comparison to baselines is poor"**
> We ran additional baselines: In the tower-building domain, we added a baseline  inspired by the model used in Bunel et al. (2018). In the list processing domain, we have added a DeepCoder model for our revision, and we are currently also running a RobustFill baseline for the list processing domain. We note that each domain now contains at least one strong baseline from previous work, in addition to the ablation models (neural semantics and RNN encoder).
>
> - **“What's the embedding function for concrete expressions?”**
> In principle, the embedding function can be any function which maps concrete expressions to vectors. The particular embedding function used is a domain-specific choice, depending on the nature of the data structure in question. From our supplement: “Tower images are embedded with a simple CNN-ReLU-MaxPool architecture, as in Ellis et al. (2020).”
> In the list processing domain: “Integers are encoded digit-wise via a GRU. Lists are encoded via a GRU encoding over the representations of the integers they contain.”
> In the string editing domain: “Input and output strings are encoded by embedding each character via a 20-dimensional character embedding and concatenating the resulting vectors to form a representation for each string.”

---

> > ### Author Response · Authors · 2020-11-19
> > **Response to Reviewer 4 (2/2)**
> >
> > - **“What's a precise definition of context? In the examples, you've shown value assignments to variables, but that doesn't allow for having multiple HOLEs in a program. Do you only allow one HOLE?”**
> > A context is defined as a binding of variables to their values, ie, {x=4, y=3}
> > In our implementation, we use a lambda calculus formulation with de Bruijn indices, so the context is given by an ordered set of values (`[4, 3]`). It is possible to have programs with multiple holes, but multiple holes with the same context will embed to the same vector. This is by design, as identical values should be represented identically.
> > - **“The policy isn't a cost function. You mean, presumably, the probability associated with the chosen action? Something else? Please explain.”**
> > Yes, the negative log probability of the chosen action is taken to be the prior cost, as described in section 3. We have updated the draft to make this more clear.
> > - **“Please explain your x axis. What are programs considered? In total? Per problem? After how much training? How big are the datasets?”**
> > Where they were missing, we have added these details to the draft---thanks for the suggestion! In our search figures, we measure the percentage of test problems solved as a function of the number of search nodes (partial programs) considered per synthesis problem. We have also added dataset size information to the main text; other training details can be found in Appendix B.
> > - **“how was test construction exactly? ‘combinations of motifs seen during training’ isn't very precise.”**
> > To construct the test set for the tower-building domain, tower objects seen during training were composed in previously unseen ways, by stacking towers on top of each other, or placing them side-by- side. We have updated this in the draft for clarity.
> > - **“Are neural modules separate for every primitive operation? Even for learned functions?”**
> > There are separate neural modules for each base function in the DSL. We do not use separate modules for the invented functions in the tower domain, as we can encode them as the composition of the modules of the primitive functions from which they are composed.
> >
> > Again, thank you for your helpful comments!
> >
> > References:
> > Hila Peleg, Roi Gabay, Shachar Itzhaky, and Eran Yahav. Programming with a read-eval-synth loop. Proc. ACM Program. Lang., (OOPSLA), 2020. URL https://cseweb.ucsd.edu/~hpeleg/resl-oopsla20.pdf.
> >
> > Bunel, Rudy, Matthew Hausknecht, Jacob Devlin, Rishabh Singh, and Pushmeet Kohli. "Leveraging Grammar and Reinforcement Learning for Neural Program Synthesis." In International Conference on Learning Representations. 2018.

---

### Official Review · AnonReviewer3 · 2020-10-26
**Interesting and novel program-synthesis technique, could use some additional experiments**

**Rating:** 7
**Confidence:** 4

**Review:**

### Summary

This paper proposes a novel top-down program synthesis for programming-by-example which combines concrete evaluation with neural embeddings. The authors take inspiration from abstract execution, which can execute partial programs by abstractly representing sets of possible execution states. Instead of hand-designing an abstract execution method, however, they propose a neural equivalent, which instead embeds possible states into a feature vector. While this approach has weaker guarantees than traditional abstract execution, it is much more flexible, and can be used as a powerful guiding function for execution-based top-down program search.

The paper is well written and was a pleasure to read. The method appears to be novel and is motivated well, and it shows strong results on a variety of program synthesis tasks, including tasks that similar previous models cannot handle. I do think that the authors should include some experiments with simpler tasks and more baselines, to give a better sense of how their method compares to prior work in the settings where that prior work is applicable. But overall, I think this paper is good and deserves to be accepted.

### Detailed comments

The paper describes three types of semantics:
- concrete semantics, where subexpressions of a (functional) program are evaluated to their concrete outputs according to the rules of a DSL
- neural semantics, where subexpressions of a program are embedded into feature vectors, and these feature vectors are transformed using DSL-based neural modules (i.e. a learned neural "sum" module is used to convert from embeddings of two integers to an embedding of their sum); this allows processing inputs that have holes in them
- blended semantics, which does a sort of partial execution: any subexpression that can be concretely evaluated is evaluated, then everything else is embedded and neural semantics are used to combine these concrete subexpressions with other partially-specified components.

Both neural and blended semantics make it possible to "execute" programs with holes in them, obtaining either concrete or neurally-embedded outputs. The authors propose using this as input to a policy $\pi$ that is trained to generate programs in the training set when conditioned on the corresponding input-output examples. They then additionally use this policy to train a value function that can be used to guide a search algorithm.

The experimental results cover three different domains: tower construction, list processing, and string editing. For the list processing domain, the authors note that previous approaches such as PCCoder cannot handle arbitrary lambda functions. The proposed blended-semantics model is able to work even in the presence of much more complex lambda functions, and the authors provide strong experimental results on an extended list-processing task with those complex lambdas. However, one thing that I think is missing is a comparison against PCCoder in the more limited domain in which both methods are applicable (the original domain without complex lambdas). Is there still value in using blended semantics in that domain, or is PCCoder better in the cases where it can be applied?

### Questions

During iterative construction of partial programs, do you allow the model to choose which hole to fill, or does the construction process require holes to be filled in some specific order (perhaps left-to-right)?

What does it mean to embed a hole "based on context"? Is the dictionary of bindings passed to a neural network in some way?

It's interesting that neural semantics can still be used even when there are concrete values. Have you considered using the concrete semantics as a regularization method for the neural semantics? For instance, by adding a loss term so that $[[f(x, y)]]^{nn} \approx [[f]]^{nn}([[x]]^{nn}, [[y ]]^{nn})$?

For end-to-end training, is the $Embed$ function trained jointly with the policy, but then fixed when training the value function? Or is there a different $Embed$ function used for the value function?

It's a bit unusual to use logistic loss when training a value function. Is this essentially a binary cross-entropy objective trained with successfull rollouts as positive examples and unsuccessful ones as negative examples?

Using "null" as the embedding of arguments to lambda expressions is an interesting choice. Does that mean that there's no way to tell which argument is which for lambdas with multiple arguments? (I also notice that appendix B.2 seems inconsistent with appendix A in this regard, stating that lambda arguments are embedded by name?)

It would be interesting to see some example programs that the method is able to produce, perhaps as part of the appendix. I'm also curious whether you have any intuition for what the neural semantics "mean" for a program with holes; have you done any analysis of this?

---

> ### Author Response · Authors · 2020-11-19
> **Response to Reviewer 3**
>
> Thank you very much for your very helpful comments and positive reception of our work! Our response is below.
>
> - **“One thing that I think is missing is a comparison against PCCoder in the more limited domain in which both methods are applicable (the original domain without complex lambdas). Is there still value in using blended semantics in that domain, or is PCCoder better in the cases where it can be applied?”**
> Conceptually, PCCoder (Zohar & Wolf, 2018) is an instance of the execution-guided synthesis family of approaches, applied to the list processing domain. In our string editing experiments (for which execution guided synthesis is possible), we showed that the REPL system---which employs a hand-coded, domain specific execution model in-the-loop of search--does achieve better test-time performance. We would expect similar results in the simple list processing domain for which PCCoder is applicable. See our response to Reviewer 4 for a short explanation as to why we expect execution-guided synthesis to outperform other approaches, but also a discussion about the conditions under which it isn’t applicable.
> - **“Do you allow the model to choose which hole to fill, or does the construction process require holes to be filled in some specific order (perhaps left-to-right)?”**
> For our experiments, we have been fixing the order in which holes are filled. For the tower and string-editing domains, which use continuation-passing style, holes are filled in left-to-right. For the list editing domain, holes are filled in right-to-left. Extensions of this work could involve models selecting which holes to fill (in which order), which may be helpful for domains in which predicting the correct next step is easier along some paths rather than others, but in a problem-specific way.
> - **“What does it mean to embed a hole ‘based on context’? Is the dictionary of bindings passed to a neural network in some way?”**
> Yes, the set of bindings is embedded by first embedding each binding and combining the embeddings via a pooling mechanism. In our implementation, we use a lambda calculus formulation with de Bruijn indices, so the context is given by an ordered set of values. We therefore pool the embeddings of each value in the context with an RNN to preserve order. The context embedding is then transformed into a hole representation by applying the hole module.
> - **“Have you considered using the concrete semantics as a regularization method for the neural semantics?”**
> We have not considered the regularization method you proposed, and think it could be very interesting! Thank you for the suggestion!
> - **“For end-to-end training, is the Embed function trained jointly with the policy, but then fixed when training the value function? Or is there a different Embed function used for the value function?”**
> Yes, in our experiments, the Embed function is trained jointly with the policy. In our experiments, the policy and value functions have entirely separate weights, including separate Embed functions.
> - **“It's a bit unusual to use logistic loss when training a value function. Is this essentially a binary cross-entropy objective trained with successful rollouts as positive examples and unsuccessful ones as negative examples?”**
> Yes, that is correct. We follow this slightly non-standard formulation from Ellis et al. (2019). This lets us interpret the value function as an estimate of the probability of successful synthesis under the policy. Although we didn’t use it in our experiments, this probabilistic interpretation would allow us to use the Sequential Monte Carlo (SMC) search procedure introduced in Ellis et al. (2019).
> - **“Using ‘null’ as the embedding of arguments to lambda expressions is an interesting choice. Does that mean that there's no way to tell which argument is which for lambdas with multiple arguments? (I also notice that appendix B.2 seems inconsistent with appendix A in this regard, stating that lambda arguments are embedded by name?)”**
> In our implementation, arguments to lambda expressions whose values have not yet been assigned at encoding time are encoded by arbitrary learned vectors. The statement in Appendix B.2 is saying that we allow the network to learn separate representations for the first unbound variable argument and the second unbound variable argument (i.e., `x` and `y`, or `$0` and `$1` in de Bruijn index notation), so that for lambdas with multiple arguments, the representation of the lambda can be sensitive to which argument position corresponds to which variable in the lambda expression. In the worst case, the network could learn to represent these vectors as “top,” but it can also learn other representations, if it is more useful.
>
> Again, thank you for your helpful review.

---

> > ### Author Response · Authors · 2020-11-19
> > **References for response to Reviewer 3**
> >
> > Ellis, Kevin, Maxwell Nye, Yewen Pu, Felix Sosa, Josh Tenenbaum, and Armando Solar-Lezama. "Write, execute, assess: Program synthesis with a repl." In Advances in Neural Information Processing Systems, pp. 9169-9178. 2019.
> >
> > Zohar, Amit, and Lior Wolf. "Automatic program synthesis of long programs with a learned garbage collector." In Advances in Neural Information Processing Systems, pp. 2094-2103. 2018.

---

> > ### Comment · AnonReviewer3 · 2020-11-20
> > **A few more suggestions / questions**
> >
> > Thank you for the clarifications.
> >
> > - **Comparison agains PCCoder**: I see, so you would expect that PCCoder would outperform your method in the simpler setting, and the value of your method is that it is more general than execution-guided approaches like PCCoder, not that it is more powerful. That seems reasonable. The new comparison against DeepCoder is perhaps more relevant, then, since DeepCoder is similarly general but underperforms your method.
> > - **Order of filling holes**: Perhaps it would be worth adding a short note in the appendix about the construction order, since that detail isn't obvious from the rest of the paper.
> > - **Embedding of lambda arguments**: That explanation makes sense, but I don't think it is consistent with the semantics you present in appendix A. I'd suggest rewording this to be clearer about what actually happens. In particular, appendix A states that the embedding of a lambda expression is the embedding of the body in an environment where the parameter $x$ is bound to null. But it sounds like $x$ should actually be "bound" to a symbolic de Bruijn index, or perhaps to a learned embedding of that symbolic de Bruijn index. The semantics in appendix A seem to require that every argument is embedded the same way (as $Embed(null)$). (Another possibility is that the embedding of identifier usages is incorrect. Perhaps $[[x]]^{nn}_{x=v}$ should be equal to $Embed(\text{'x'})$ or $Embed([\text{'x'}, v])$ and not just $Embed(v)$?
> >
> >   A related question on this topic: do you ever apply neural semantics in a context where variables have values? Your semantics doesn't seem to have any way to extend the context with anything other than $null$. (Does the context start with values in it?)

---

> > > ### Author Response · Authors · 2020-11-21
> > > **Response to additional suggestions and questions**
> > >
> > > - **Construction order:**
> > > Thank you for the suggestion regarding construction order. We have added a note about this to Appendix B.
> > > - **Embedding of lambda arguments:**
> > > Thank you for this suggestion. We’ve updated the semantics in Appendix A, so it is clearer and more consistent. To account for the case when variables are assigned to null in the context, we have added the rule:
> > > $$
> > > [[x]]^{nn}\_{C |= x=null} = Embed (\text{'x'})
> > > $$
> > > where $ Embed(\text{'x'}) $ returns a representation corresponding to the variable $x$ (each variable has a separate learned embedding). When variables are assigned to a value $v$ that is not null, we apply the original rule:
> > > $$
> > > [[x]]^{nn}_{C |= x=v} = Embed(v)
> > > $$
> > > - **“Do you ever apply neural semantics in a context where variables have values?”** Yes, we do. For example, for synthesis from input-output examples, the context is initialized with the inputs, and these values can be applied when the corresponding variable appears in the program. As an example, if the spec contains the example 3 -> 7, then the context will be initialized with the input value: context = {x = 3}. Wherever the program contains the variable assigned to the input, the input value will be substituted in place of that variable (in this case, `x`). (This is the example depicted in Section 3.)
> > >
> > > Again, thank you for your very helpful comments. Let us know if there are any remaining questions.

---

### Official Review · AnonReviewer2 · 2020-10-28
**Nice technique but incomplete semantics, and unclear scope and generalizability**

**Rating:** 6
**Confidence:** 4

**Review:**

This paper proposes an embedding mechanism for partial programs for search space exploration in example-driven synthesis. It executes a sub-expression concretely whenever possible and applies neural module networks on vector representations otherwise. The embeddings of partial programs and goal states are used for determining the next step towards expanding an unfilled hole. This method is evaluated on three benchmark sets: tower construction, functional list processing and string editing.

The idea of using module networks to embed partial program states is nice and these embeddings are blended with concrete state representations. This combination is shown to perform better than concrete execution-guided synthesis or embedding program syntax (through an RNN).

I am not clear about the scope of this work, and whether it will generalize beyond these specific domains and bechmarks. The paper presents concrete and abstract denotational semantics. The examples and (some) benchmarks include loops. However, the semantics do not formalize the iterative semantics (in the concrete case) or the fixpoint semantics (in the abstract case). The training proceeds by imitation learning on the syntactic search space. Thus, the proposed method does not seem to address the inductiveness of loop invariants. A 'loop' is probably treated as any other function. How will this method then generalize to different loop bounds? It is possible that the training set contains enough examples for it to memorize patterns necessary for the test set. A more convincing evaluation would be to have benchmarks where training is restricted up to loop bounds, say k, and the test set contains programs with loop bounds m > k.

The functional list processing benchmark uses higher-order functions. The abstract interpretation of such programs requires reasoning about relational semantics. How are these semantics formalized and encoded by neural networks? In what ways can we expect the neural network to generalize?

The other problem is that of potential non-termination. It is easy to synthesis a non-terminating program even with finite loop bounds, for instance, if the loop counter is decremented in the loop body and never hits the loop termination condition. This is a problem that can make the synthesis procedure itself non-terminating, if such a loop is synthesized and is executed concretely. An abstract interpreter can also get into non-termination unless suitable operations (such as widening) are employed. The paper should talk about this issue and how it can be mitigated.

The semantics are not fully formalized. What is the initial state of a program? Are the values undefined or are they initialized to some default constants? This has a bearing on what can be executed concretely and what cannot be. The paper says 'buildColumn(1)' can be executed concretely since all its arguments are concrete. But doesn't it implicitly need the position of the cursor where the column should be built? I suppose the initial state of the cursor is assumed to be set to 0, which makes this a concrete statement.

The discussion about comparison to abstract interpretation needs more clarity. In particular, how the results of abstract interpretation are used should be stated. One might use a Hoare style reasoning to check whether the invariant implies the post-condition (outputs). However, here it is the opposite direction for one wants to check "realizability" of the partial program.

Please cite the original paper on abstract interpretation: "Abstract interpretation: a unified lattice model for static analysis of programs by construction or approximation of fixpoints" (POPL'77).

There is a typo in Sec 3. 'y =9' should be 'y = 7'.

---

> ### Author Response · Authors · 2020-11-19
> **Response to Reviewer 2**
>
> Thank you very much for your helpful comments. Our response is below.
>
> - Your review raises important questions about how loops and loop invariants are handled by our method. You are correct that our approach handles loops generically, the same way as any other function. This means the neural module has to compute a representation of a looping program using a fixed, feed-forward computation, without an iterative fixpoint computation. (This also allows us to avoid issues of non-termination.) It’s possible that a more rigorous treatment of loops could lead to improved accuracy, though it would introduce additional complexity to the method.
> - **“The semantics are not fully formalized”**
> We have added details to clarify the semantics of the tower-building domain in the appendix. You are correct in assuming that the initial state of the program has the canvas empty, the cursor at location 0 (the middle of the canvas), and the cursor orientation set to 1 (towards the right). Please let us know if more details would be helpful.
> - We have updated the discussion of the comparison to abstract interpretation in the draft. In particular, we have made it more clear that in the “Comparison to abstract interpretation” experiment, abstract interpretation is employed by checking whether the desired output state is within the abstract state given by executing the partial program, and rejecting candidate partial programs for which this does not hold.  We have also updated the related work section to highlight previous approaches which utilize abstract interpretation to determine if a desired state is reachable from a given partial program.
>
> Details:
> - We have added a citation to the original abstract interpretation paper.
> - Thanks for picking up that typo! We’ve fixed it.
>
> Once again, thank you for your thoughtful review.

---

> > ### Comment · AnonReviewer2 · 2020-11-21
> > **Clarify limitations**
> >
> > Thank you responding to my comments and updating the paper.
> >
> > I couldn't find formalization of semantics of higher-order functions and discussion of how the network learns those. It would help the reader to have these details.
> >
> > I strongly recommend that you add a limitations section to discuss the simplistic non-inductive treatment of loops, and your handling of higher-order functions.

---

> > > ### Author Response · Authors · 2020-11-22
> > > **RE: Clarify limitations**
> > >
> > > Thank you very much for your response.
> > >
> > > - **Limitations section:**
> > > Thank you for the suggestion regarding a limitations section. We have written a limitations section discussing the treatment of loops and lambda functions, which can be found in the appendix (and is referenced in Section 3).
> > >
> > > - **"I couldn't find formalization of semantics of higher-order functions and discussion of how the network learns those. It would help the reader to have these details."**
> > > The neural semantics of higher-order functions are the same as those of first-order functions, in that the function is represented by a neural module, and each argument is embedded and treated as a vector input to the neural module. See the definition of $[[E]]^{nn}_C$ in Section 3 (and Appendix A):
> > > $$[[f(E_1 … )]]^{nn}_C = [[f]]^{nn}([[E_1]]^{nn}_C …)$$
> > > A more detailed discussion of the neural semantics of lambdas and functions (which includes higher-order functions) can be found in Appendix A. In both Section 3 and Appendix A, we have added text to make it clear that the discussion of neural semantics applies to both first-order and higher-order functions. If this is not what you were referring to, please let us know what details should be clarified, and we will be happy to provide whatever additional information would be helpful.
> > >
> > > Again, thank you for your helpful comments.

---

> > > > ### Author Response · Authors · 2020-11-23
> > > > **Limitations**
> > > >
> > > > We have just updated the draft again with a clearer pointer to the limitations section, and we agree that this is a very important addition to the paper. We believe that exploring these questions is a promising direction for future work, and we appreciate you highlighting these points.

---

> > > > > ### Comment · AnonReviewer2 · 2020-11-24
> > > > > **Concerns addressed**
> > > > >
> > > > > Thank you for including more detailed semantics and discussing the limitations. This addresses my concerns.

---

### Official Review · AnonReviewer1 · 2020-10-30
**(Updated) Review of Representing Partial Programs with Blended Abstract Semantics**

**Rating:** 7
**Confidence:** 4

**Review:**

Update:

The authors have addressed many of my comments below. As such, I am increasing my score.

High-level view:

I’m slightly negative on this paper. The ideas within it seem novel to me, but the experimental examination seems unnecessarily weak / limited. I think if the authors perform a more rigorous analysis of their system, this will result in a strong tier-1 publication. In its current form, I cannot argue for its acceptance at ICLR.


Summary:

This paper introduces an interesting concept of using blended abstract semantics to assist with program synthesis (PS). As I understand it, PS is the field of research interested automatically building software programs using some external guidance, often times in the form of a what could be considered as the intention being conveyed by the human to the machine. Intention here derives from the nomenclature created in the “Three Pillars of Machine Programming” paper, Gottschlich et al. MAPL ‘18. These intentions may be presented many ways. For classical PS, intention tends to be expressed in the form of input / output examples, a specific set of rules, etc.

The “blended” part of the semantics seems to be a fusion of both: (i) neural abstract semantics and (ii) blended abstract semantics.

In this paper, the authors note that the recent prior work in program synthesis using traditional formal methods as well as non-traditional techniques such as machine learning are still in the early stages of exploration and can be improved upon. In particular, the authors attempt to achieve this by using a novel form of neurosymbolic representation (a type of representation that includes both a neural network reasoning as well as more classical symbolic reasoning) with their blended semantics approach. Based on the current knowledge I possess, it seems like this is an interesting and novel way to approach the problem.

The authors evaluate their system in three experimental domains: (i) tower building (think Tetris), (ii) list processing (think numerical list transformation [1, 12, 4] -> [1, 4, 12] (sort)), and (iii) string-editing (think automatic spelling correction or acronymizing things like “programming languages to PL” and “machine learning to ML”).

Overall, they show their blended semantics approach outperforms other neural systems in some cases and not others.

1.	For the tower building examples, they compare against what I would consider to be a fairly weak baseline of research systems – only two of which are not their variants of their own blended semantics and the results favor the blended semantics variants by upwards of ~29% (just a guess by staring at Figure 4).

2.	For the list processing, they compare blended semantics against an RNN and neural semantics. It seems blended semantics performs slightly better than both other systems by upwards of around 10% or so.

3.	For string editing, they compare against more well-known state-of-the-art synthesizers like RobustFill (ICML 2017) and REPL (NeurIPS 2019). However, they note that REPL outperforms blended semantics, by a fairly large margin for text editing. In the best case, it seems REPL is ~25% more correct than blended semantics; in the worst case, it seems REPL is 60x better. However, it’s a bit hard to tell because the authors have overlaid the legend on *top* of the REPL curve in Figure 7, so one can only speculate about certain aspects there. I’d kindly request that in the next version of the paper, the graph legend in Figure 7 be moved to a place such that it’s not blocking the readers view of the actual performance we are trying to analyze.

Overall, I don’t think this is a bad paper, but I think it has too many weaknesses in its current form for me to give it a positive rating. I discuss some of these below.


High-level concerns:

With respect to the empirical evaluation, I agree that there are likely many important problems in the space of list processing and string editing. Things like sort for list processing is ubiquitous as is string editing for automatic spell checking. So, I have no problems with understanding the validity of these domains and their examination.

However, the tower building experimental domain left me wanting more. More in at least two dimensions. First, I wanted more state-of-the-art systems to consider against. I don’t think a vanilla RNN is really sufficient as a reasonable baseline. Perhaps I’m missing something and the authors can explain to me why I might be wrong? Second, I don’t understand “why I should care” about tower building problems. I don’t feel the problem domain was properly motivated (or maybe I just missed it).

Lastly, these experiments seem shallow to me compared to other program synthesis work I’ve seen at NeurIPS and ICLR. Let’s take Hoppity (ICLR 2020), which looked something like 300k JavaScript bugs and was able to synthesize solutions to around 10,000 of them out of something like 36k. Keep in mind these were real bugs, in a real world programming language.

I realize using an embedded DSL for classic PS has different types of limitations, but just for the sake of comparison I found the experiments (and the motivation of the “hole” example in the tower) a bit underwhelming. And yes, I realize that Armando Solar-Lezama’s sketch system also addressed holes – but if you read his PhD dissertation those holes were with actual code (as I understood it), not necessarily a single instead of a Tetris-like game.

I also was pretty disappointed by the related work section. Armando Solar-Lezama has done incredible work in machine programming for the last 15 years. However, he’s not the only one publishing in the space. Five of the 21 references are of Armando’s work. That’s 24% of the citations coming from one person. The authors also introduce the field of program synthesis from Armando’s PhD dissertation on sketching. Please read the “Three Pillars of Machine Programming” (who Armando also co-authored) and note how those others refer to the space of machine programming by citing an original work in the 1950s by IBM on the future of Fortran. *This* in my opinion, is the paper to cite when you are saying this:
“Synthesizing programs from examples is a classic AI problem which has seen advances from the Pro- gramming Languages community (Solar-Lezama, 2008).”
A citation to a 2008 paper doesn’t demonstrate (at least to me) anything classic about the problem. But a 1950s paper, which means we’ve been working on it for 70 years probably would.

Moreover, in the related work section, there’s a bunch of citations to work in the last 5-10 years, which is great, but most don’t seem to provide any intuition about how these systems work, except those on execution-guided neural program synthesis. I’d recommend this section be rewritten with greater care to provide at least a basic intuition about how each of these related works relate


Low-level concerns:

Please define all of your acronyms before you use them. I counted at least these: RNN, GRN, MDP, MSE, DSL, REPL. This is especially important because there are three disjoint communities being brought together here: the programming languages community, formal methods community, and the machine learning community. I think it’s unlikely except for experts in all three domains to implicitly know that DSL mean domain specific language and simultaneously knowing GRU means gated recurrent unit.

---

> ### Author Response · Authors · 2020-11-19
> **Response to Reviewer 1**
>
> Thank you very much for your helpful and constructive comments. We have incorporated your comments into our draft, and we respond below:
> - We have added additional baselines for our experiments. In the tower domain, our new baseline uses a CNN to encode the specification image, and an LSTM decoder decodes the tokens of the target program while attending spatially over the image features from the CNN. This model is inspired by the model used in Bunel et al. (2018) for the image-based Karel synthesis domain, with the addition of spatial attention. We have also added a DeepCoder baseline for the list processing domain, and are additionally running a RobustFill baseline for list processing. With these additional baselines, each domain now contains at least one strong model from previous work, in addition to the ablation models (neural semantics and RNN encoder).
> - We appreciate the suggestions for how to improve the related work section. We have used the additional space to significantly expand this section, adding more detailed discussion of connections between this work and other research topics in the program synthesis literature. We have added a more representative and diverse set of citations from the neural program synthesis space, and provided more clarity and detail concerning how the cited papers relate to the current work.
> - We have incorporated additional discussion motivating the tower-building domain. At a high level, we are interested in program synthesis not only for assisting software development, but also because we see program synthesis as an essential tool for general AI. The tower-building domain is inspired by and builds upon classic work in AI (Winston, 1972). It is also related to important problems in the AI literature, such as generalized planning, where plans can be represented as programs (Jimenez et al., 2018), and often require looping constructs (Srivastava et al., 2015). We also note that the tower-building domain was used in Ellis et al. (2020) as a target domain for library-learning techniques. In this work, our DSL includes library functions automatically learned by the techniques in Ellis et al. (2020). Our experiments in this domain therefore explore the compatibility of our approach with DSL library-learning techniques. We hope that our additional discussion clarifies this point, and helps add context to the use of the tower-building domain.
> - **“Lastly, these experiments seem shallow to me compared to other program synthesis work I’ve seen at NeurIPS and ICLR.”**
> We would like to emphasize an important distinction between some of the referenced work (e.g. Hoppity, ICLR 2020) and the current paper. The referenced work focuses on learning **code -> code** transformations (e.g. fixing bugs in already-written code). This is a different problem from the **specification -> code** task studied in our submission. While approaches for learning code transformations are often evaluated on large datasets of real code, they do not synthesize entire programs from scratch given specifications. For synthesizing entire programs from examples, our experimental domains are comparable to (and in several cases the same as) state-of-the-art work on program synthesis, see Ellis et al., 2019 (NeurIPS 2019); Chen et al., 2018 (ICLR 2019); Zohar & Wolf, 2018 (NeurIPS 2018); Bunel et al., 2018 (ICLR 2018).
>
> Minor comments:
> - We added transparency to the legend for the string-editing graph for better readability.
> - We made sure to clearly define acronyms before using them, thanks for pointing this out.
>
> Again, we really appreciate the detailed review and hope that the new draft addresses your concerns.

---

> > ### Author Response · Authors · 2020-11-19
> > **References for response to Reviewer 1**
> >
> > Jiménez, Sergio, Javier Segovia-Aguas, and Anders Jonsson. "A review of generalized planning." The Knowledge Engineering Review 34 (2019): e5.
> >
> > Srivastava, Siddharth, Shlomo Zilberstein, Abhishek Gupta, Pieter Abbeel, and Stuart Russell. "Tractability of planning with loops." In Twenty-Ninth AAAI Conference on Artificial Intelligence. 2015.
> >
> > Winston, Patrick. “The MIT robot.” Machine Intelligence, 1972
> >
> > Devlin, Jacob, Jonathan Uesato, Surya Bhupatiraju, Rishabh Singh, Abdel-rahman Mohamed, and Pushmeet Kohli. "RobustFill: Neural Program Learning under Noisy I/O." In International Conference on Machine Learning, pp. 990-998. 2017.
> >
> > Zohar, Amit, and Lior Wolf. "Automatic program synthesis of long programs with a learned garbage collector." In Advances in Neural Information Processing Systems, pp. 2094-2103. 2018.
> >
> > Chen, Xinyun, Chang Liu, and Dawn Song. "Execution-guided neural program synthesis." In International Conference on Learning Representations. 2018
> >
> > Ellis, Kevin, Maxwell Nye, Yewen Pu, Felix Sosa, Josh Tenenbaum, and Armando Solar-Lezama. "Write, execute, assess: Program synthesis with a repl." In Advances in Neural Information Processing Systems, pp. 9169-9178. 2019..
> >
> > Ellis, Kevin, Catherine Wong, Maxwell Nye, Mathias Sable-Meyer, Luc Cary, Lucas Morales, Luke Hewitt, Armando Solar-Lezama, and Joshua B. Tenenbaum. "Dreamcoder: Growing generalizable, interpretable knowledge with wake-sleep bayesian program learning." arXiv preprint arXiv:2006.08381 (2020).
> >
> > Bunel, Rudy, Matthew Hausknecht, Jacob Devlin, Rishabh Singh, and Pushmeet Kohli. "Leveraging Grammar and Reinforcement Learning for Neural Program Synthesis." In International Conference on Learning Representations. 2018.

---

> > ### Comment · AnonReviewer1 · 2020-11-23
> > **I'll increase my score**
> >
> > Thanks for the detailed response and changes to your paper to address my outstanding concerns. I've reviewed the new version of the paper and I think it now warrants an increase in score of 6 or 7 (for me). Given that it seems I was previously the most negative of this paper, I will no longer stand in the way of this paper being accepted.
> >
> > I appreciate you taking the time to address my concerns.

---

### Author Response · Authors · 2020-11-19
**Summary of changes**

We would like to thank all of the reviewers for their very helpful and constructive comments.

We appreciated the positive comments on the submission: “The idea of using module networks to embed partial program states is nice” (R2) “Blended neural semantics is an elegant and intuitive construct.” (R4) “The paper is well written and was a pleasure to read. The method appears to be novel and is motivated well, and it shows strong results on a variety of program synthesis tasks, including tasks that similar previous models cannot handle … overall, I think this paper is good and deserves to be accepted.” (R3)

We’ve uploaded a revised draft incorporating reviewer feedback. Below is a summary of the main changes:
- (R1, R3, R4) We have added two additional baselines for our experiments, so that each domain now contains at least one neural synthesis baseline from the literature, in addition to the ablation models (neural semantics and RNN encoder). In the tower-building domain, our new baseline is inspired by the model used in Bunel et al. (2018) for the image-based Karel synthesis domain, with the addition of spatial attention. We have also added a DeepCoder baseline for the list processing domain. Neither of these new baselines meets the performance of our proposed method. We are additionally running a RobustFill baseline for the list processing domain; we will update the draft when this experiment is completed.
- (R1) We have revised the related work section, to more clearly discuss connections between this work and a number of other research topics in the program synthesis literature, including neurally-guided search techniques, neural program representation, execution-guided synthesis, and synthesis via abstraction interpretation.
- (R1, R2, R3, R4) We have added various model, domain and evaluation clarifications, as discussed in responses to individual reviews.
- (R2, R4) We have added more complete semantics for the tower-building domain.
- (R2) We have added a section entitled "Limitations of our neural semantics" to the appendix.

References:
Bunel, Rudy, Matthew Hausknecht, Jacob Devlin, Rishabh Singh, and Pushmeet Kohli. "Leveraging Grammar and Reinforcement Learning for Neural Program Synthesis." In International Conference on Learning Representations. 2018.

---

### Decision · Program_Chairs · 2021-01-07
**Final Decision**

**Decision:**

Accept (Poster)

**Comment:**

Four expert reviewers (after much discussion, in which the authors seemed to do a pretty good job addressing a lot of the initial complaints) unanimously voted to accept this paper.

Everyone seemed to agree that the idea was interesting, and it is indeed interesting.
There were generally complaints about benchmarking; there always are for papers about program synthesis.

One complaint I have, but that I didn't really see mentioned, is that the system as described is pretty baroque.
I have a hard time imaging how you'd scale something like this up to more complicated contexts,
and honestly I'm not sure even in some of the contexts where it was tested if it would really outperform a well-engineered
top-down synthesizer.
Maybe this is just an aesthetic preference that only I have, and maybe ideas need to start out overly complicated
before the most useful bits can be extracted from them and refined.

At any rate, I do think that this paper gives a cool new research contribution and that people will want to read it, so I am recommending acceptance.